# Impact of Chronic Kidney Disease and Dialysis on Outcome after Surgery for Infective Endocarditis

**DOI:** 10.3390/jcm12185948

**Published:** 2023-09-13

**Authors:** Ahmed Elderia, Ellen Kiehn, Ilija Djordjevic, Stephen Gerfer, Kaveh Eghbalzadeh, Christopher Gaisendrees, Antje-Christin Deppe, Elmar Kuhn, Thorsten Wahlers, Carolyn Weber

**Affiliations:** Department of Cardiothoracic Surgery, Heart Center, University of Cologne, 50937 Köln, Germany; ellen.kiehn@uk-koeln.de (E.K.); ilija.djordjevic@uk-koeln.de (I.D.); stephen.gerfer@uk-koeln.de (S.G.); kaveh.eghbalzadeh@uk-koeln.de (K.E.); christopher.gaisendrees@uk-koeln.de (C.G.); antje-christin.deppe@uk-koeln.de (A.-C.D.); elmar.kuhn@uk-koeln.de (E.K.); thorsten.wahlers@uk-koeln.de (T.W.)

**Keywords:** infective endocarditis, chronic renal failure, chronic kidney disease, renal insufficiency, dialysis, valve surgery

## Abstract

Infective endocarditis (IE) carries a heavy burden of morbidity and mortality in chronic kidney disease (CKD) and hemodialysis (HD) patients. We investigated the risk factors, pathognomonic profile and outcomes of surgically treated IE in CKD and HD patients. We preoperatively identified patients with CKD under hemodialysis (HD group) and compared them with patients without hemodialysis (Non-HD group). Furthermore, we divided the cohort into four groups according to the underlying stage of CKD, with a subsequent outcome analysis. Between 2009 and 2018, 534 Non-HD and 58 HD patients underwent surgery for IE at our institution. The median age was 65.1 [50.6–73.6] and 63.2 [53.4–72.8] years in the Non-HD and HD groups, respectively (*p* = 0.861). The median EuroSCORE II was 8.0 [5.0–10.0] vs. 9.5 [7.0–12.0] in the Non-HD vs. HD groups (*p* = 0.004). Patients without CKD had a mortality rate of 5.6% at 30 days and 15.5% at 1 year. Mortality rates proportionally rose with the severity of CKD. Among HD patients, 30-day and 1-year mortality rates were 38.1% and 75.6%, respectively (*p* < 0.001). *Staphylococcus aureus* IE was significantly more frequent in the HD group (*p* = 0.006). In conclusion, outcomes after surgery for IE correlated with the severity of the underlying CKD, with HD patients exhibiting the most unfavorable results. Pre-existing CKD and staphylococcus aureus infection were independent risk factors for 1-year mortality.

## 1. Introduction

Patients with advanced chronic kidney disease (CKD) suffer metabolic, immune and inflammatory disorders [1]. Cardiovascular complications followed by infections are the leading mortality causes in CKD patients. Infective endocarditis (IE), as both a cardiac and infectious disease, carries a heavy burden of morbidity and mortality in these patients. With an incidence of 2–6%, CKD patients under hemodialysis (HD) experience IE at a risk 40–60 times greater than the general population [2,3]. Repeated venous access in HD patients might allow virulent organisms to enter the circulation, with the tricuspid valve being the first contact point. Nevertheless, a literature review reported a considerably higher prevalence of left-sided IE in HD patients, varying from 80% to 100%, than right-sided IE, which accounted for 0% to 26% [2]. In our analysis, we examined the characteristics and outcomes of patients undergoing surgery for IE, especially HD patients. Our aims were to (1) describe HD patients’ preoperative data, comorbidities and the spectrum of pathogens in IE; (2) analyze the perioperative results; (3) investigate whether postoperative outcomes, especially survival, preoperatively correlated with the severity of underlying CKD; and (4) identify independent predictors of 30-day mortality, 1-year mortality and newly required dialysis after surgery for IE.

## 2. Materials and Methods

### 2.1. Study Population

This was a monocentric observational analysis of consecutive patients surgically treated for IE between January 2009 and December 2018. Acquired data included patients’ demographics, predisposing risk factors, echocardiographic and microbiologic findings, perioperative data and relevant clinical outcomes. Long-term follow-up was obtained by reviewing hospital medical records and interviewing patients, their relatives or patients’ physicians. The study protocol was approved by the institutional ethics committees (Ethics Committee of the Medical Faculty, University of Cologne, 17-407).

### 2.2. Study Design

For the subsequent analyses, the patient population was divided into two groups: patients without a preoperative need for hemodialysis (Non-HD) and patients with CKD requiring preoperative hemodialysis (HD group). An additional analysis was conducted by categorizing the population into four groups based on their preoperative glomerular filtration rates (GFR). This allowed us to perform a more detailed examination of the patients’ conditions and analyze the outcomes accordingly.

### 2.3. Definition of Variables

Infective endocarditis was defined according to the recent modified Duke criteria. Surgery was indicated in compliance with current European Society of Cardiology guidelines [4]. CKD was defined and classified according to the KDIGO classification (Kidney Disease—Improving Global Outcomes) and KDIGO 2012 guidelines [5]. Acute kidney injury (AKI) was defined according to the Kidney International Supplements of 2012 [6]. The age of patients was recorded at the time of hospital admission due to IE. The EuroSCORE II was calculated in order to estimate the perioperative mortality risk. Immunosuppression was defined as the administration of immunosuppressive medication, including chemotherapy, corticosteroids (at any dosage) or drugs that target the immune system as part of antirheumatic therapy. Known previous cerebrovascular events (CVEs) were defined as ischemic or hemorrhagic cerebral insults in relation to IE. Postoperative CVEs were considered to be any new-onset neurologic deficits with a cerebral origin occurring during the primary hospital stay in association with signs of hemorrhage or ischemia on CT/MRI scans of the brain, along with a clinical assessment by a neurologist. The 30-day and 1-year mortality included death from any cause within the first 30 days and between day 31 and day 365 after surgery, respectively. Late mortality was defined as all-cause mortality occurring during the follow-up period. The follow-up time for survival was measured from the date of operation to either the date of death or the date of the last contact with the patient.

### 2.4. Management of Patients

The clinical condition, echocardiographic and microbiologic findings of each patient were evaluated with appropriate assessment of the operative risk and timing of the procedure. An interdisciplinary endocarditis team consensually chose the antimicrobial regime and duration of therapy according to recent guidelines. Surgery for IE was indicated according to the recent European Society of Cardiology guidelines for the management of infective endocarditis [4]. All operations were performed under general anesthesia via median sternotomy with a routine establishment of cardiopulmonary bypass (CPB) techniques utilizing roller-head pumps, a membrane oxygenator, cardiotomy suction, moderate systemic hypothermia (34 °C) and cardioplegic arrest. The selection of the surgical approach and the type of valve prosthesis used were decided by the attending cardiac surgeon taking into account factors such as the severity of complication associated with IE, anatomical considerations and the patient’s age and preferences.

### 2.5. Statistical Analysis

All data were statistically analyzed using SPSS^®^ Statistics version 28.0 (IBM Corporation, Armonk, New York, NY, USA). For continuous variables, depending on the distribution, the mean or the median were given with the respective standard deviation or the 25th and 75th percentile. Group comparisons were performed using an unpaired Student’s *t*-test or Mann–Whitney U test. Discrete variables were expressed as percentages and tested with a chi-squared test or Fisher’s exact test. Missing data were not imputed and were randomly assumed to be missing. The 30-day (day 1–30) and 1-year mortality (day 31–365) were reported, and univariable and multivariable analyses were performed for risk stratification. Logistic regression was used to identify possible predictors for 30-day mortality and Cox regression was used for 1-year mortality. After excluding patients with preoperative dialysis, logistic regression was used to adjust for differences between groups with possible confounding factors and to assess possible predictors for new-onset postoperative dialysis. After the univariable analysis, all variables with a *p*-value of less than 0.1 were entered into the multivariable model using a forward selection (likelihood ratio; *p* = 0.05). The results were presented as the odds ratio (OR) or hazard ratio (HR) with a corresponding 95% confidence interval (CI) and *p*-value. All reported *p*-values were two-sided and were considered to be statistically significant if ≤5%. In addition, we performed a Kaplan–Meier analysis for long-term survival.

## 3. Results

Data of 592 patients who underwent surgery due to IE were retrospectively analyzed. The median age of patients in the whole cohort was 64.7 years [50.9–73.5]. The majority of patients were male (74.8%). In total, 143 (26.9%) patients preoperatively presented with normal kidney functions and were classified into the G1 group. Underlying preoperative CKD, from G2 to G5, was identified in 398 patients (74.8%). Preoperative HD, the G5 group, was identified in 58 (9.8%) patients. Figure 1 displays the distribution of patients according to the CKD stage. The first comparison was conducted between Non-HD patients (*n* = 534) and HD patients (*n* = 58). A second analysis according to CKD stage followed. The median duration of the follow-up was 2.4 years (interquartile range (IQR) 0.26–5.8), with a completeness of 76.3%.

### 3.1. Comparison between Non-HD and HD Patients

The median age of the Non-HD patients was 65.1 [50.6–73.6] vs. 63.2 [53.4–63.2] years for HD patients (*p* = 0.861). The median EuroSCORE II in the Non-HD group was 8.0 [5.0–10.0], significantly lower than the HD group at 9.5 [7.0–12.0] (*p* = 0.004). Except for diabetes mellitus, there were no significant differences regarding pre-existing conditions and comorbidities, as listed in Table 1.

Among the predisposing factors for developing IE, immunosuppression was documented in only 1.5% of Non-HD patients compared with 6.9% of HD patients (*p* = 0.024). Other risk factors, including a previous history of IE, valvular or congenital heart disease and intravenous drug abuse, did not differ between groups (see Appendix A, for risk factors among HD and Non-HD patients).

In the entire patient population, left-heart endocarditis (94.2%) was significantly more common than right-heart endocarditis (6.7%). Double-valve endocarditis occurred in 13.5% (*n* = 72) in the Non-HD group and slightly more frequently in the HD group at 19.0% (*n* = 11), yet this difference was not significant (*p* = 0.259) (see Appendix A, for the distribution of IE-affected valves among HD and Non-HD patients).

In HD patients, the aortic valve was less frequently affected than the mitral valve at 46.5% vs. 62.0% (*p* = 0.046 and *p* = 0.022), respectively (see Figure 2, which illustrates the frequencies of IE-affected valves in HD and Non-HD patients). In the whole cohort, endocarditis of a native valve (NVE) was more common (77.6%) than endocarditis of a valve prosthesis (PVE) (22.3%). PVE of the mitral valve was significantly more frequent in HD patients at 12.1% vs. 3.8% in the Non-HD group (*p* = 0.014).

IE caused by staphylococcus aureus was significantly more frequent in the HD group (*p* = 0.006) (the microbiological findings of both groups are listed in Appendix A).

Cross-clamp, cardiopulmonary bypass and overall operation time were significantly longer in the HD group (see Appendix A, for the operative data of the examined cohort).

Non-HD patients remained in the intensive care unit (ICU) for a shorter time-period than HD patients (5.0 [2.0–9.0] and 8.0 [4.0–14.0] days, respectively (*p* < 0.001)). There was no significant difference in the length of hospital stay between groups; yet, in the course of follow-up, hospital re-admission was significantly more frequent in the HD group. Postoperative tracheotomy was performed in 13.7% and 31.0% of Non-HD and HD patients (*p* < 0.001), respectively. CVEs occurred more frequently in the HD group (12.1% vs. 4.9% (*p* = 0.046)). The postoperative outcomes are listed in Table 2.

In the whole cohort, 19.4% died 30 days after surgery; 17.4% in the Non-HD group and 38.1% in the HD group (*p* = 0.001). One year after surgery, 33.7% of the entire group had died; 29.1% in the Non-HD group and 75.6% in the HD group (*p* < 0.001). HD patients had a median survival of only 1.6 [0.8–2.4] years. The survival probabilities of HD and Non-HD patients are shown in Figure 3. Among all causes of death in deceased patients, only sepsis was significantly more common in HD patients (*p* < 0.001). The causes of death in the examined population are listed in Appendix A.

### 3.2. Results Based on CKD Stage

As the HD group (corresponding with G5) was examined in detail in the previous section, the focus of the subsequent analyses is on the groups with mild–moderate (G2/G3a) and moderate–severe (G3b/G4) kidney disease without the need for HD. There were no significant differences between groups regarding the preoperative left ventricular ejection fraction (LVEF). Of the patients in the G1 group, 7.6% had LVEF < 30% vs. 2.9% in the G2/G3a group (*p* = 0.285) and 6.7% in the G3b/G4 group (*p* = 0.942). Regarding postoperative outcomes, patients without preoperative CKD (G1) had significantly fewer re-thoracotomy and tracheotomy procedures than patients with CKD in both groups. The difference between groups was more pronounced in the moderate–severe (G3b/G4) group than in the mild–moderate (G2/G3a) group (see Table 3).

The evaluation of postoperative creatinine levels showed a significance correlating with the underlying stage of CKD (*p* < 0.001 for each) (see Appendix B, Figure A1).

The 30-day mortality and 1-year mortality showed a highly significant difference correlating with the stage of CKD (*p* < 0.001) (see Appendix A). Survival according to CKD stage is graphically shown in Figure 4. The lowest percentage of mortality was always G1 patients (5.6% 30-day mortality and 15.5% 1-year mortality), and mortality increased with an advancing stage of CKD. Consequently, G5 patients had the highest mortality, with a 10-fold increase in 30-day mortality and 5-fold increase in 1-year mortality compared with patients without CKD (53.8% and 75.0%, respectively) (*p* < 0.001).

### 3.3. Predictors for 30-Day Mortality, 1-Year Mortality and New Postoperative Dialysis

The multivariable analysis revealed the following variables as independent predictors for 30-day and 1-year mortality: age over 65 years, male sex, coronary artery disease, preoperative CKD and infection with staphylococcus aureus (see Table 4). Independent predictors for new postoperative dialysis were an age over 65 years, peripheral vascular disease, preoperative CKD and PVE (see Table 5).

## 4. Discussion

HD patients operated on for IE had concomitant diabetes mellitus and mitral valve involvement more often and staphylococcus aureus was the most frequently detected causative organism. The 30-day mortality in G5 patients was 10 times higher than G1 patients. Three-quarters (75%) of HD patients died within the first year. CKD patient survival was significantly shorter the more severe the pre-existing CKD stage was. Even mild–moderate CKD led to a 2–3-fold increase in 30-day and 1-year mortality. Additionally, the more severe the stage of CKD, the more frequent the postoperative complications like re-thoracotomy and tracheotomy.

### 4.1. Demographics and Intraoperative Course

The median age of the entire population was 64.7 [50.9–73.5] years and the median age of HD patients was 63.2 years [53.4–72.8]. This was slightly higher than the median age of other study collectives [7]. Of the 592 patients, 441 were male (74.5%) and 149 (25.2%) were female. The study revealed a higher affection of men than women by IE, as also shown in the literature [7,8]. The significantly higher EuroSCORE II in the HD group was a correlate of marked multimorbidity. Diabetes mellitus is a common comorbidity in HD patients, as observed in the examined population and in other collectives [9,10]. As diabetic nephropathy is a frequent complication of long-standing diabetes mellitus, there could be a causal relationship. The increased mortality could also be related to both diabetes mellitus and preoperative renal insufficiency, as both illnesses can severely impair the general condition of a patient and can represent a decisive factor for the weakened immune system in severe diseases such as IE. Unlike our results, Chaudry et al. found that diabetes mellitus was an independent risk factor for elevated 1-year mortality in dialysis patients undergoing surgery for IE [10].

The reason for the significantly longer duration of the operation (246 vs. 202 min), time on CPB (140 vs. 114 min) and ischemia time (96 vs. 74 min) in HD patients remains unclear. Omoto et al. reported a more frequent occurrence of perivalvular abscesses in dialysis patients who underwent surgery for IE [11]. Presumed higher rates of peri-/intraoperative complications (abscess, fistula or perforation) in HD patients were not found in the collective examined here.

The fact that the mitral valve is the most frequently affected valve in patients with preoperative dialysis has also been observed in other study groups [12,13]. Mitral valve prosthesis endocarditis has also been observed significantly more frequently in dialysis patients. In patients without dialysis, the aortic valve is the most frequently affected valve. It is likely that other means of infection, especially more nosocomial infections due to dialysis and an altered immune defense, could be a cause of this difference. A legitimate explanation cannot currently be given with certainty.

### 4.2. Spectrum of Pathogens

Staphylococcus aureus is often detected in connection with medical interventions or nosocomial infections [14]. Similar to our results, many studies have reported that staphylococcus aureus was the most common pathogen causing IE in patients requiring HD [7,13,14]. A possible causal relationship may be due to the installation of an invasive access point for dialysis, which poses a risk of bacteremia and subsequent infection. This would also correlate with regard to the high mortality of endocarditis patients with preoperative dialysis as staphylococcus aureus is a very virulent organism, thus promoting a fulminant infection and a worse outcome [15]. Streptococci were less frequently detected in our HD patients. This finding was consistent with the results of a study by Bhatia et al. [7].

### 4.3. Postoperative Course and Complications

The ICU and in-hospital length of stay were significantly longer in the HD group. The 30-day, 1-year and long-term mortality were significantly higher in HD patients, with sepsis the most frequent cause of death. Comparable with our data, significantly higher mortality rates among patients with IE requiring dialysis compared with patients without preoperative dialysis have been established in numerous studies [11,12,16]. A recent meta-analysis revealed an overall in-hospital death rate of 29.5% (95% CI: 26.7–46.6%) and long-term death rate of 45.6% (95% CI: 41.9–49.3%) in HD patients suffering infective endocarditis [17]. A large prospective cohort study of 6691 patients diagnosed with IE showed that both in-hospital and 6-month mortality rates were significantly higher in HD patients (8.3% of the whole cohort) compared with Non-HD patients at 30.4% vs. 17% and 39.8% and 20.7%, respectively [18]. In a retrospective population-based cohort analysis form Scotland, Gallacher et al. reported that patients hospitalized for IE who were undergoing dialysis or who had undergone kidney transplantation exhibited increased odds of 1-year (45%) and 3-year mortality (64%) when compared with a group without advanced CKD (31% and 43%, respectively) [19]. A prospective analysis from a Cleveland clinic showed a 5-year survival of only 24% after surgery for IE in HD patients, which was substantially worse than in Non-HD patients (about 65%). Despite this, the rate was only 15% worse than the survival of the general HD population without IE; it was substantially better than that for HD patients with non-surgically treated IE (only 11%) [20].

Postoperative CVEs were more frequent in the HD group at 12.1% vs. 4.9% in the Non-HD group (*p* = 0.046). Likewise, Bhatia et al. reported an increased occurrence of cerebrovascular complications in dialysis patients undergoing surgery for IE [7]. A study by Misfeld et al., which examined IE patients with cerebral events, identified the bypass time during surgery and preoperative dialysis as independent risk factors for worse outcomes [21].

### 4.4. Independent Predictors for 30-Day and 1-Year Mortality and New-Onset Postoperative Dialysis

An age over 65 years, the male gender, coronary artery disease, preoperative CKD and infection with staphylococcus aureus were significantly correlated with 30-day and 1-year mortality. In a similar study population, age and diabetes mellitus were found to be independent risk factors for 1-year mortality [10]. Murdoch et al. also found that an older age, staphylococcus aureus infection and PVE increased the risk of in-hospital mortality [22]. A recent retrospective analysis reported that the use of central venous catheters and inadequate antimicrobial management increased mortality due to IE [16].

PVE, nephrotoxic medication, an advanced age, diabetes mellitus and CKD are known independent predictors for new-onset dialysis in patients with IE [23,24]. Liu et al. was able to identify streptococci infections as an independent risk factor for postoperative dialysis, which was not observed in our cohort [25]. Consistent with our results, however, was that postoperative new-onset dialysis was significantly more common in the group of patients with pre-existing kidney disease from stage G2a onwards. Obviously, a pre-existing kidney disease increases vulnerability to the progression of kidney damage implicated by IE, with consequently higher mortality [25,26].

Despite comprehensive medical and surgical care, IE patients, particularly those with pre-existing CKD, still have a very high mortality rate. This implies that further research and improvement in prevention, diagnostics and therapy are essential for this patient collective. In line, this study helps to better characterize patients with IE and preoperative CKD as this data allow for a better identification of and subsequent preventive measures for patients at risk.

### 4.5. Limitations

The collected data originated from an unselected cohort from a single center. Although we presented a relatively large collective of surgically treated IE patients, compared with the available literature, the analysis may have been strengthened by including a larger number of patients and conducting a multicenter study. A higher completeness of follow-ups would have strengthened our statement. Some variables had too many missing data and had to be removed from the analysis (e.g., etiology and duration underlying CKD and immunosuppression as well as preoperative NYHA class). Unfortunately, the GFR values were only available for 532/592 patients (10% missing) and for only 22 of the 58 patients requiring preoperative dialysis, so the G5 collective was significantly smaller in the evaluation according to GFR stages. As the HD group (corresponding with G5) was examined in detail in the first analysis, the focus of the second analysis, according to GFR, was on the patient groups without vs. patients with mild–moderate (G2/G3a) and moderate–severe (G3b/G4) kidney disease without HD.

## 5. Conclusions

HD patients had significantly higher mortality after surgery for IE. Our analysis showed a correlation between the severity of the underlying CKD stage and postoperative outcome after surgery for IE. Pre-existing CKD, PVE and staphylococcus aureus infection were identified as independent risk factors for 1-year mortality.

## Figures and Tables

**Figure 1 jcm-12-05948-f001:**
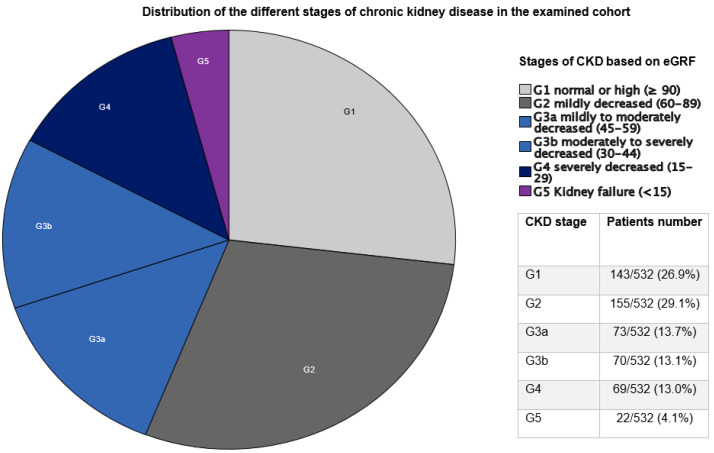
Distribution of the different stages of chronic kidney disease in the examined cohort. CKD: chronic kidney disease; eGFR: estimated glomerular filtration rate; G1 group: patients without CKD; G2/G3a groups: patients with mild–moderate CKD; G3b/G4 groups: patients with moderate–severe CKD; G5 group: patients with kidney failure.

**Figure 2 jcm-12-05948-f002:**
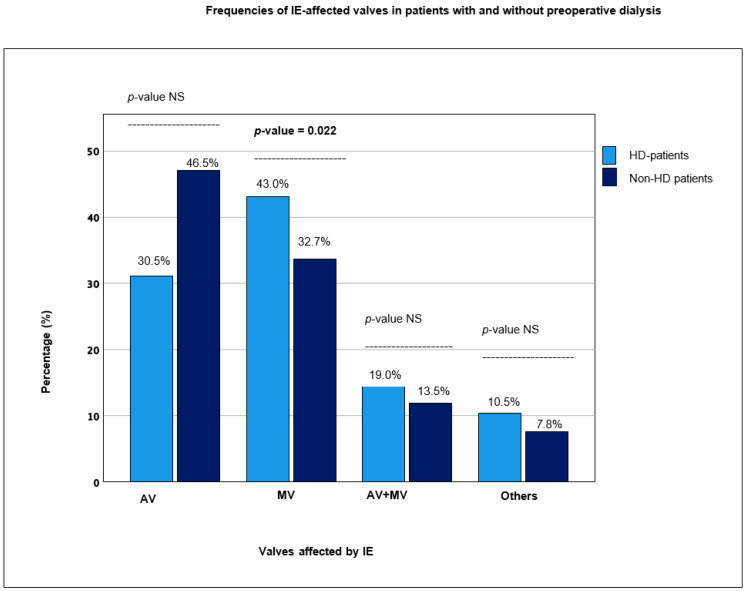
Frequencies of IE-affected valves in patients with and without preoperative dialysis. HD: hemodialysis; IE: infective endocarditis; NS: non-significant; AV: IE involving aortic valve; MV: IE involving mitral valve; AV + MV: IE involving both aortic and mitral valves; Others: pulmonary and/or tricuspid valves.

**Figure 3 jcm-12-05948-f003:**
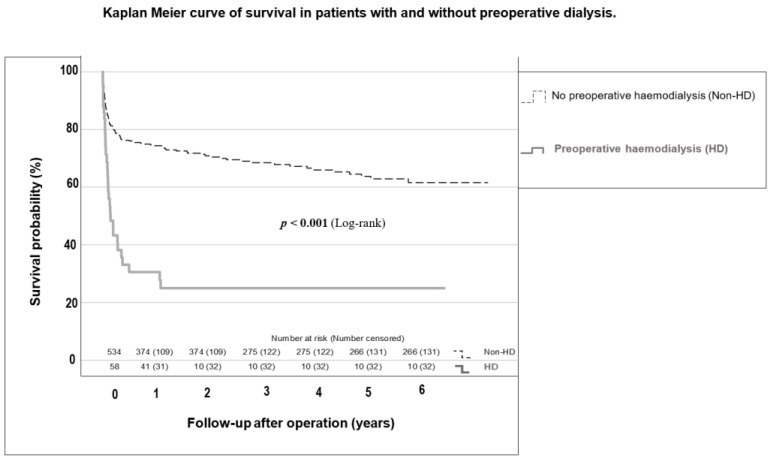
Kaplan–Meier curve of survival of patients with and without preoperative dialysis. HD group: patients with preoperative hemodialysis; Non-HD group: patients without preoperative hemodialysis.

**Figure 4 jcm-12-05948-f004:**
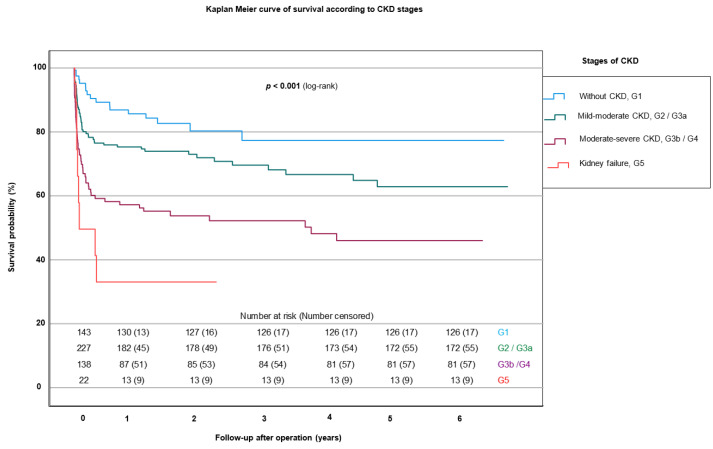
Kaplan–Meier curves of survival according to CKD stage. CKD: chronic kidney disease; G1 group: patients without CKD; G2/G3a groups: patients with mild–moderate CKD; G3b/G4 groups: patients with moderate–severe CKD; G5 group: patients with kidney failure.

**Table 1 jcm-12-05948-t001:** Characteristics of patients surgically treated for infective endocarditis.

Variables	All Patients (*n* = 592)	Non-HD Group (*n* = 534)	HD Group (*n* = 58)	*p*-Value
Age (years)	64.7 [50.9–73.5] ^a^	65.1 [50.6–73.6] ^a^	63.2 [53.4–72.8] ^a^	0.861
Female	149 (25.2%)	132 (24.9%)	17 (29.3%)	0.464
BMI (kg/m^2^)	25.5 [23.3–28.7] ^a^	25.5 [23.3–28.7] ^a^	25.0 [23.5–28.8] ^a^	0.953
COPD	60 (10.2%)	53 (10.0%)	7 (12.1%)	0.614
Art. HTN	374 (63.4%)	331 (62.2%)	43 (74.1%)	0.074
PHT	54 (9.2%)	46 (8.6%)	8 (13.8%)	0.197
Hyperlipidemia	176 (29.8%)	159 (29.9%)	17 (29.3%)	0.927
CAD	169 (28.6%)	147 (27.6%)	22 (37.9%)	0.099
PVD	55 (9.3%)	46 (8.6%)	9 (15.5%)	0.087
Diabetes mellitus	153 (25.9%)	127 (23.9%)	26 (44.8%)	**<0.001**
Known CVEs	76 (12.9%)	69 (13.0%)	7 (12.1%)	0.846
Active smoker	125 (21.2%)	114 (21.4%)	11 (19.0%)	0.663
LVEF < 30%	15 (2.6%)	14 (2.7%)	1 (1.7%)	0.666
LVEF 30–50%	108 (26.6%)	92 (26.3%)	16 (28.5%)	
LVEF > 50%	274 (67.6%)	236 (67.6%)	38 (67.8%)	
EuroSCORE II	8.0 [6.0–11.0] ^a^	8.0 [5.0–10.0] ^a^	9.5 [7.0–12.0] ^a^	**0.004**

Metric, non-normally distributed variables were calculated as the median with 25th and 75th percentiles (^a^). For nominal variables, the absolute number (*n*) was calculated with the percentage (%). **Bold** indicates *p* < 0.05. Non-HD group: patients without preoperative hemodialysis; HD group: patients with preoperative hemodialysis; BMI: body mass index; COPD: chronic obstructive pulmonary disease; Art. HTN: arterial hypertension; PHT: pulmonary hypertension; CAD: coronary artery disease; PVD: peripheral vascular disease; CVEs: cerebrovascular events; LVEF: left ventricular ejection fraction.

**Table 2 jcm-12-05948-t002:** Postoperative outcomes of patients undergoing surgery for infective endocarditis.

Variables	All Patients *n* = 592	Non-HD Group *n* = 534	HD Group *n* = 58	*p*-Value
Re-thoracotomy	93 (15.8%)	83 (15.7%)	10 (17.2%)	0.754
Tracheotomy	91 (15.4%)	73 (13.7%)	18 (31.0%)	**<0.001**
ICU stay (days)	5.0 [2.0–10.0] ^a^	5.0 [2.0–9.0] ^a^	8.0 [4.0–14.0] ^a^	**<0.001**
In-hospital stay (days)	13.0 [9.0–8.0] ^a^	13.0 [9.0–18.0] ^a^	11.0 [8.0–19.0] ^a^	0.193
New pacemaker implantation	59 (10.0%)	56 (10.5%)	3 (5.2%)	0.196
Myocardial infarction	5 (0.8%)	4 (0.8%)	1 (1.7%)	0.497
Cerebrovascular events	33 (5.6%)	26 (4.9%)	7 (12.1%)	**0.046**
Hospital re-admission	176 (53.0%)	158 (51.3%)	18 (75.0%)	**0.022**
IE recurrence	24 (7.3%)	22 (7.2%)	2 (8.7%)	0.797
30-day mortality	85 (19.4%)	69 (17.4%)	16 (38.1%)	**0.001**
1-year mortality	140 (33.7%)	109 (29.1%)	31 (75.6%)	**<0.001**
Median survival (days)	4.3 [4.0–4.6] ^a^	4.5 [4.2–4.8] ^a^	1.6 [0.8–2.4] ^a^	**<0.001**

Metric, non-normally distributed variables were calculated as the median with 25th and 75th percentiles (^a^). For nominal variables, the absolute number (*n*) was calculated with the percentage (%). **Bold** indicates *p* < 0.05. Non-HD group: patients without preoperative hemodialysis; HD group: patients with preoperative hemodialysis; ICU: intensive care unit; IE: infective endocarditis.

**Table 3 jcm-12-05948-t003:** Frequencies of postoperative complications according to CKD stage.

Variables	G1 Group *n* = 143 (26.9%)	G2/G3a Groups *n* = 228 (42.8%)	*p*-Value	G3b/G4 Groups *n* = 139 (26.1%)	*p*-Value
Postoperative AKI	21/143 (14.7%)	70/226 (31.0%)	**<0.001**	70/139 (50.4%)	**<0.001**
Re-thoracotomy	11/143 (7.7%)	38/226 (16.8%)	**0.012**	31/139 (22.3%)	**<0.001**
Tracheotomy	10/143 (7.0%)	33/227 (14.5%)	**0.027**	33/138 (23.9%)	**<0.001**
Postoperative coronary angiography	1/142 (0.7%)	2/227 (0.9%)	0.854	0/138 (0.0%)	0.243
Myocardial infarction	2/142 (1.4%)	1/227 (0.4%)	0.322	1/139 (0.7%)	0.574
New pacemaker implantation	10/143 (7.0%)	23/227 (10.1%)	0.302	16/139 (11.5%)	0.190
CVEs	5/143 (3.5%)	11/226 (4.9%)	0.529	12/138 (8.7%)	0.068
IE recurrence	6/71 (8.5%)	8/140 (5.7%)	0.458	4/71 (5.6%)	0.512
Hospital re-admission	33/72 (45.8%)	72/140 (51.4%)	0.440	39/73 (53.4%)	0.361

CKD: chronic kidney disease; G1 group: patients without CKD; G2/G3a groups: patients with mild–moderate CKD; G3b/G4 groups: patients with moderate–severe CKD; AKI: acute kidney injury; CVEs: cerebrovascular events; IE: infective endocarditis. **Bold** indicates *p* < 0.05.

**Table 4 jcm-12-05948-t004:** Independent predictors for 30-day and 1-year mortality among patients operated on for infective endocarditis.

30-Day Mortality			1-Year Mortality		
Preoperative Risk Factor	OR [95% CI]	*p*-Value	Preoperative Risk Factor	OR [95% CI]	*p*-Value
Age > 65 years	1.929 [1.192–3.121]	**0.007**	Age > 65 years	1.955 [1.339–2.854]	**<0.001**
Male gender	1.924 [1.212–3.054]	**0.006**	Male gender	1.809 [1.316–2.714]	**<0.001**
Coronary artery disease	1.655 [1.054–2.596]	**0.028**	Coronary artery disease	1.590 [1.118–2.261]	**0.010**
Preoperative renal insufficiency	2.729 [1.641–4.537]	**<0.001**	Preoperative renal insufficiency	2.493 [1.684–3.691]	**<0.001**
Staphylococcus aureus infection	1.747 [1.106–2.761]	**0.017**	Staphylococcus aureus infection	1.567 [1.080–2.273]	**0.018**
Perivalvular abscess	1.725 [1.108–2.684]	**0.016**	Prosthetic valve endocarditis	1.578 [1.102–2.259]	**0.013**

**Bold** indicates *p* < 0.05; OR: odds ratio; CI: confidence interval.

**Table 5 jcm-12-05948-t005:** Independent predictors for new-onset requirement for hemodialysis in patients operated on for infective endocarditis.

Variables	OR [95% CI]	*p*-Value
Age > 65 years	2.458 [1.346–4.490]	**0.003**
Peripheral vascular disease	1.679 [1.034–2.728]	**0.036**
Preoperative renal insufficiency	4.134 [2.245–7.612]	**<0.001**
Prosthetic valve endocarditis	2.062 [1.225–3.469]	**0.006**

**Bold** indicates *p* < 0.05; OR: odds ratio; CI: confidence interval.

## Data Availability

The data that support the findings of this study are available from the corresponding authors (A.E and C.W.) upon reasonable request.

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
