# Peer review of "Impact of Chronic Kidney Disease and Dialysis on Outcome after Surgery for Infective Endocarditis"

_jcm, 2023, doi:10.3390/jcm12185948_

Round 1

Reviewer 1 Report

This is a single center observational analysis of consecutive patients treated surgically for IE aiming to determine the impact of preoperative kidney function on hard outcomes and describe relevant clinical features.  Patient population was initially divided into two groups: patients without need for haemodialysis preoperatively and patients with CKD requiring haemodialysis preoperatively. An additional analysis by categorizing the population into four groups based on their preoperative glomerular filtration rates was performed to provide a more detailed examination of the patients' conditions and analyse the outcomes accordingly. The study is largely derivative and confirmative of available knowledge in the field and suffer from the usual limitations of retrospective analyses. A major strength is the large sample despite a consistent rate of patients lost to follow up. All in all, beside the above mentioned, there are no major flaws in methods and presentation. The conclusions, though supported by the data, are those expected in almost all the subsets of patients referred for cardiac surgery. The manuscript is well written but the clinical and scientific bottom line is predictable and without a clear appeal for the readership. 

Author Response

Comments to the Authors: Point 1: The study is largely derivative and confirmative of available knowledge in the field and suffer from the usual limitations of retrospective analyses.

Reply to point 1: We thank the reviewer for the time and effort he invested in reviewing our manuscript. We greatly appreciate the critical and constructive comments. We provide a contemporary study with a relatively large IE sample size for a single center analysis. We agree with the reviewer, that others have also described the clinical characteristics and outcomes of patients with CKD undergoing surgery for IE. However, the reason for the considerably higher risk of patients with CKD undergoing surgery for IE is still not understood and guidelines based on prospective randomized trials for the best treatment for CKD patients with IE are still lacking. 

Comments to the Authors: Point 2: The manuscript is well written but the clinical and scientific bottom line is predictable and without a clear appeal for the readership.

Reply to point 2: We thank the reviewer for commenting on this important circumstance. According to the reviewer’s helpful recommendations and comments, we revised our manuscript in order to improve the focus of the manuscript. We now focus more on the underlying stadium of CKD influencing mortality in patients undergoing surgery for IE. We do not only report on dialysis yes/no, but also analyse the different stages of CKD and can show that even mild renal dysfunction increases the risk of mortality. Therefore, we can give the clinician a message that even patients with mild CKD need to be meticulously monitored. In the case of the application of potentially nephrotoxic substances (antibiotics, analgesics), in particular, the administration must be carefully considered and, if necessary, the dose must be adapted to the renal function.

Reviewer 2 Report

This is a well written manuscript discussing the impact of advanced CKD on perioperative outcomes in a cohort of infective endocarditis patients. There are several limitations that must be addressed.

Major comments

1) Methods are subject to several biases. First, no power calculations have been applied. Second, data are not available for about 24% of the patients screened, which is a significant dropout rate, Third, why a p value <0.1 was set as the level to include a variable as a potential predictor in multivariate analysis ? Fourth, only surgically treated patients were included which could have affected outcomes. Surgically treated IE cases are morbid patients with increased incidence of adverse outcomes. On the contrary the vast majority of IE cases are treated medically. What was the rationale of opting for surgically managed cases of IE?

2) The etiology of CKD should be adequately presented in the manuscript. Furthermore it should be explained if renal function was deteriorated due to the infection/ antibiotics/ hemodynamics during hospitalization. For instance were there any case of acute kidney injury on top of CKD or de novo AKI? These data are valubale and should be meticulously described, if available.

Minor Comments

1) Data regarding NYHA class, LVEF and urgency of operation must be provided, since they are indicative of Euroscore and clinical presentation

2) Immunosuppressive conditions should be further explained and precisely described in the methodoloy. There is no universal definition and you have to be more elaborate on this topic. For instance cortisol therapy dosage and duration, chemo, advanced cancer etc should be described

3) table 2 median survival is expressed in days!! I think you ment years. Please correct it.

English are fine

Author Response

Comments to the authors: Point 1: Methods are subject to several biases. First, no power calculations have been applied. 

Reply to point 1: We thank the reviewer for the time and effort he invested in reviewing our manuscript. We greatly appreciate the critical and constructive comments. And indeed, we did not perform a power calculation in this paper as the main results, especially survival, are quite evident. Moreover, post-hoc power analysis has been criticized as a means of interpreting negative study results. Because post-hoc analyses are typically only calculated on negative trials (p ≥ 0.05), such an analysis will produce a low post-hoc power result, which may be misinterpreted as the trial having inadequate power.*

* Levine M, Ensom MH. Post hoc power analysis: an idea whose time has passed? Pharmacotherapy: The Journal of Human Pharmacology and Drug Therapy 2001; 21(4): 405-9.

Comments to the authors: Point 2: Second, data are not available for about 24% of the patients screened, which is a significant dropout rate

Reply to point 2: We thank the reviewer for commenting on this circumstance. We agree with the reviewer, that a higher completeness of follow-ups would have strengthened our manuscript. In case of a registry, 23.6% of missing follow-ups is not optimal but seems adequate. We added this point to the limitations of our paper. In order to improve the reliability of our data, our aim is to conduct a multicenter prospective endocarditis registry.

Comments to the authors: Point 3: Third, why a p value <0.1 was set as the level to include a variable as a potential predictor in multivariate analysis?

Reply to point 3: We thank the reviewer for commenting on this point. In consultation with our local statisticians, we did the following: after univariate analysis all variables with a p-value less than 0.1 were entered into the multivariable model using a forward selection (likelihood ratio, pin = 0.05). Applied multivariable regression analysis with forward selection was validated using additional multivariable regression model with backward selection (likelihood ratio, pout = 0.1). Only if both methods led to the same conclusion the results were expected to be reliable. After consulting our statistician and double checking, the number of independent variables in the multivariable model seems adequate for the number of events.

Comments to the authors: Point 4: Fourth, only surgically treated patients were included which could have affected outcomes. What was the rationale of opting for surgically managed cases of IE?

Reply to point 4: We thank the reviewer for commenting on this circumstance. We agree with the reviewer that the inclusion of only surgically treated endocarditis patients is a limitation. Unfortunately, because we provide our data of a cardiac surgery department, data from patients receiving conservative therapy are not available. Although we cannot make a statement about the entire endocarditis collective (conservatively and surgically treated), we can show many aspects in this work that could be interesting for a broad readership.

Comments to the authors: Point 5: The etiology of CKD should be adequately presented in the manuscript. 

Reply to point 5: We thank the reviewer for commenting on this circumstance. We agree with the reviewer that information on etiology of CKD would be of great interest to JCM's readership. Unfortunately we have not recorded this for all patients and therefore cannot provide it. Due to the urgency of the surgery and the deteriorated clinical status of most patients, it was not possible to collect this data. But we think that with the GFR we have an objectifiable parameter that can give a reliable statement about the renal function of examined population.

Comments to the authors: Point 6: For instance were there any case of acute kidney injury on top of CKD or de novo AKI? These data are valuable and should be meticulously described, if available.

Reply to point 6: We thank the reviewer for commenting on this point. We reported the frequencies of AKI according to the preoperative stage of CKD in table 3 in the main manuscript. 26.9% had de novo AKI, while 42.8% of CKD patients in stages G2/G3a and 26.1% in stages G3b/G4 had AKI on top of CKD. Furthermore, we added a diagram of creatinine levels course postoperatively as a supplemental figure (Supplemental Figure A1). Moreover, we performed a univariable and multivariable regression analysis to identify the independent predictors for new postoperative dialysis. These were initially submitted as a supplemental table (Supplemental Table S7). In the revised Manuscript, we added the relevant independent predictors for new postoperative dialysis as table 5.

Table 3. Frequencies of postoperative complications according to CKD stage.

variables

G1 group

n=143

(26.9%)

G2+G3a

n=228

(42.8%)

p-value

G3b+G4

n=139

(26.1%)

p-value

postoperative AKI

21/143

(14.7%)

70/226

(31.0%)

<0.001

70/139

(50.4%)

<0.001

Table 5: Independent predictors for new-onset requirement for haemodialysis in patients operated for infective endocarditis.

variables

Odds Ratio

[95% confidence interval]

p-value

Age > 65 years

2.458 [1.346 – 4.490]

0.003

Peripheral vascular disease

1.679 [1.034 – 2.728]

0.036

Preoperative renal insufficiency

4.134 [2.245 – 7.612]

< 0.001

Prosthetic valve endocarditis

2.062 [1.225 – 3.469]

0.006

Age > 65 years

2.458 [1.346 – 4.490]

0.003

Comments to the authors: Point 7: Data regarding NYHA class, LVEF and urgency of operation must be provided, since they are indicative of EuroScore and clinical presentation.

Reply to point 7: We thank the reviewer for commenting on this point. We agree with the reviewer that information on preoperative symptoms, urgency and LVEF would be of interest to JCM's readership. Unfortunately we have not recorded NYHA class for all patients and therefore cannot provide data on it. Since it is often an emergency operation in which no detailed anamnesis could be collected due to the urgency of the operation and sometimes intubated/non-contactable patients. Therefore there were too many missing data and we had to remove these variables from our database. Nevertheless, we will consider this constructive suggestion for further prospective analyses. We indicated this as a limitation in the revised manuscript: ‘’ some variables had too many missing data and had to be removed from the analysis (e.g. etiology and duration underlying CKD and immunosuppression as well as NYHA class preoperatively).''

Regarding LVEF, 2.6% of the examined population had LVEF<30% with 2.7% the Non-HD and 1.7% in the HD-group (p = 0.666). This data was added to table 1. In the comparison according to GFR stage, similar results were noticed. As there were no significant differences between groups and as LVEF is integrated in the calculation of EuroScore, we waived this data from the main manuscript in order to avoid redundancy.

Table 1. Characteristics of patients surgically treated for Infective Endocarditis.

Variables

All patients

(n=592)

Non-HD group

(n=534)

HD-group

(n=58)

p-value

LVEF<30%

15 (2.6%)

14 (2.7%)

1 (1.7%)

0.666

Comments to the authors: Point 8: Immunosuppressive conditions should be further explained and precisely described in the methodology.

Reply to point 8: We thank the reviewer for commenting on this circumstance. We added a definition to the methodology section of the revised manuscript ‘’Immunosuppression was defined as the administration of immunosuppressive medications in any dosage including chemotherapy, corticosteroids or drugs that target the immune system as part of anti-rheumatic or immunophilins therapy.’’

In the examined cohort, 12 (2.0 %) of patients had at least one of the above mentioned substances at admission. However, we cannot specify the exact underlying condition and duration of therapy for all patients, since it is often an emergency operation in which no detailed anamnesis could be collected due to the urgency of the operation and sometimes intubated/non-contactable patients. We indicated this as a limitation: ‘’ some variables had too many missing data and had to be removed from the analysis (e.g. etiology and duration underlying CKD and immunosuppression as well as NYHA class preoperatively).''

Comments to the authors: Point 9: table 2 median survival is expressed in days!! I think you ment years.

Reply to point 9: We thank the reviewer for this comment. We corrected this in the revised manuscript.

Reviewer 3 Report

The article presents a study with somewhat expected results. Nevertheles,  any data regarding the prognosis of HD/CKD patients undergoing surgery for IE  are welcome for practice.

The introduction is properly written. The methodology is properly presented. The statistical method is appropiate.

The results contain some inconsistencies, which have to be treated.

For instance:

Fig 1 contains 532 patients, the nonHD cohort consists of 534 patients? The 22 non HD G5 patients are missing from statistics (table 3).

A detailed analysis of etiological agents is missing.

Discussion - could be more results-oriented.

General observation: the English has to be improved substantially, there are many not proper terms used,

e.g. line 87 cardiologic arrest, girth of surgical remediation,

       line 160 staph-aureus (please use one form of the name)

     line 318 patient collective, etc.

has to be improved

Author Response

Comments to the authors: Point 1: Fig 1 contains 532 patients, the non-HD cohort consists of 534 patients? The 22 non HD G5 patients are missing from statistics (table 3).

Reply to point 1: We thank the reviewer for the time and effort he invested in reviewing our manuscript. We greatly appreciate the critical and constructive comments. In our Analysis, we examined Data of 592 patients who underwent surgery due to IE. The cohort was divided according to the need for dialysis preoperatively, so that the first comparison was done between Non-HD patients (n=534) and HD-patients (n=58). We conducted an additional analysis by categorizing the population according to their preoperative glomerular filtration rates (GFR). Hereby, 143 (26.9%) patients had normal kidney functions preoperatively and were classified into G1-group. An underlying preoperative CKD, from stage G2 to G5, was identified in 398 patients (74.8%) as Figure 1 displays. Preoperative HD, corresponding to G5-stage, was identified in 58 (9.8%) of patients. Unfortunately, the GFR value was available for 532/592 patients (10% missing) and only 22 of 58 patients requiring dialysis preoperatively, so that G5 collective was significantly smaller in the evaluation according to GFR stages. In addition, stage G5 patients undergoing renal replacement therapy no longer have reliably measurable GFR, which is why it would not make sense to determine their postoperative renal function or to include it in the analyses. Since the group of dialysis patients (corresponds to G5) was examined in detail in the first analyses (Non-HD vs. HD), the focus of the second analyses (according to GFR) was on the patient groups without CKD vs. patients with mild, moderate or severe CKD without HD. We therefore added this point to the limitations of our study, marked changes in the revised manuscript and indicated it below in red colour for your convenience.

 ‘’Unfortunately, the GFR values were available for 532/592 patients (10% missing) and for only 22 of 58 patients requiring dialysis preoperatively, so that the G5 collective was significantly smaller in the evaluation according to GFR stages. Since the HD-group (corresponding to G5) was examined in detail in the first analysis, the focus of the second analyses, according to GFR, was on the patient groups without vs. patients with with mild-moderate (G2/G3a) and moderate-severe (G3b/G4) kidney disease without HD.’’

Comments to the authors: Point 2: A detailed analysis of etiological agents is missing.

Reply to point 2: We thank the reviewer for commenting on this Point. We listed all relevant microbiological findings in both Non-HD and HD groups in supplemental table S3 and added it below for your convenience. IE caused by staph-aureus was significantly more frequent in the HD-group (p=0.006). We structured the discussion section and discussed these findings as below.

‘’ Staphylococcus aureus are often detected in connection with medical interventions or nosocomial infections.1 Similar to our results, many studies reported that, staphylococcus aureus was the most common pathogen causing IE in patients requiring dialysis.1-3 A possible causal relationship is the installation of an invasive access for dialysis, which poses a risk of bacteraemia and subsequent infection. This would also fit into the picture with regard to the high mortality of endocarditis patients with preoperative dialysis, since Staphylococcus aureus is a very virulent organism and thus promotes a fulminant infection and a worse outcome.4 Streptococci were less frequently detected in HD-patients. This finding is consistent with the results of a study by Bhatia et al.2 ’’

  1. Leither MD, Shroff GR, Ding S, Gilbertson DT, Herzog CA. Long-term survival of dialysis patients with bacterial endocarditis undergoing valvular replacement surgery in the United States. Circulation 2013; 128(4): 344-51.
  2. Bhatia N, Agrawal S, Garg A, et al. Trends and outcomes of infective endocarditis in patients on dialysis. Clinical cardiology 2017; 40(7): 423-9.
  3. Kamalakannan D, Pai RM, Johnson LB, Gardin JM, Saravolatz LD. Epidemiology and clinical outcomes of infective endocarditis in hemodialysis patients. The Annals of thoracic surgery 2007; 83(6): 2081-6.
  4. Luehr M, Weber C, Misfeld M, et al. Virulence of Staphylococcus infection in surgically treated patients with endocarditis: a multicenter analysis. Annals of Surgery 2023; 277(6): e1364-e72.

Supplemental Table S3: Microbiological findings in patients with and without preoperative dialysis

Variables a

All patients

n=592

Non-HD group

n=534

HD-group

n=58

p-value

Staphylococcus aureus

147 (24.9%)

124 (23.3%)

23 (39.7%)

0.006

CoNS

75 (12.7%)

69 (13.0%)

6 (10.3%)

0.569

Streptococcus pyogenes

15 (9.6%)

13 (2.4%)

2 (3.4%)

0.659

Streptococcus viridans

106 (17.9%)

99 (18.6%)

7 (12.1%)

0.218

Other Streptococci

43 (7.2%)

42 (7.9%)

1 (1.7%)

0.044

Enterococci

84 (14.2%)

79 (14.8%)

5 (8.6%)

0.197

Gram-negative HACEK

5 (0.8%)

4 (0.8%)

1 (1.7%)

0.494

Gram-negative non-HACEK

21 (3.5%)

16 (3.0%)

5 (8.6%)

0.057

Fungi

8 (1.3%)

7 (1.3%)

1 (1.7%)

0.806

Other Organisms

55 (9.3%)

53 (8.9%)

3 (5.2%)

0.397

No organisms detected

88 (14.9%)

79 (14.8%)

9 (15.5%)

0.892

Multiple infections

57 (9.6%)

52 (9.7%)

5 (8.6%)

0.723

For listed nominal variables a, the absolute number, n, is calculated with percentage (%). Bold indicates P < 0.05.

Non-HD group: patients without Haemodialysis preoperatively. HD-group: patients with Haemodialysis preoperatively. IE: infective endocarditis, CoNS: coagulase-negative staphylococci. HACEK: Haemophilus species, Aggregatibacter species, Cardiobacterium hominis, Eikenella corrodens and Kingella species.

Comments to the authors: Point 3: Discussion - could be more results-oriented.

Reply to point 3: We thank the reviewer for commenting on this point. According to the reviewer’s helpful recommendation, we revised our manuscript in order to improve the focus of the manuscript, to highlight our results as well as to be able to give clinician a clear message that even patients with mild CKD are at higher short and long-term mortality risk after surgery for IE. The lethality rises significantly with advanced CKD. Therefore, CKD patients need to be meticulously monitored preioperatively, e.g. in case of application of potentially nephrotoxic substances (antibiotics, analgesics), in particular, the administration must be carefully considered and, if necessary, the dose must be adapted to the renal function.

Comments to the authors: Point 4: the English has to be improved substantially, there are many not proper terms used, e.g. line 87 cardiologic arrest, girth of surgical remediation, line 160 staph-aureus (please use one form of the name), line 318 patient collective, etc.

Reply to point 4: We thank the reviewer for commenting on this circumstance. We adjusted the above mentioned terms and rechecked the revised manuscript by 2 co-authors, who are fluent in English writing. For your convenience, all revised terms have been marked in the revised manuscript with track changes and are indicated below in red colour.

cardiologic arrest --> cardioplegic arrest

girth of surgical remediation --> The selection of the surgical approach

staph-aureus --> staphylococcus aureus

patient collective --> cohort.

Round 2

Reviewer 2 Report

The authors timely addressed all comments. Several of them remain unsolved and hence should be considered as inherent methodological limitations. It is my opinion that more data should be available in order to convey solid clinical messages even from a retrospective registry. For example,  an urgent operation (within days) provides sufficient time to collect valuable clinical data such as etiology of CKD or LVEF. Thank you for considering my suggestions. 

None 

Author Response

Comments to the Authors: Point 1: It is my opinion that more data should be available in order to convey solid clinical messages even from a retrospective registry. For example, an urgent operation (within days) provides sufficient time to collect valuable clinical data such as etiology of CKD or LVEF.

Reply to point 1: We thank the reviewer for commenting on this circumstance. We agree with the reviewer that information on etiology of CKD would be of great interest to JCM's readership. In the examined Cohort, many patients with CKD had diabetic nephropathy and chronic glomerulonephritis. Diabetes mellitus is a common comorbidity in HD-patients as observed in our examined population and in other studies. Since diabetic nephropathy is a frequent complication of long-standing diabetes mellitus, there could be a causal relationship here. The increased mortality could also be related to both diabetes mellitus and preoperative renal insufficiency, since both previous illnesses can severely impair patient’s general condition and can represent a decisive factor for the weakened immune system in severe diseases such as IE. However, in our multivariable analysis diabetes mellitus was not an independent risk factor for elevated 1-year mortality in dialysis patients undergoing surgery for IE. Unfortunately we could not recorded exact etiological backgrounds of renal insufficiency for all patients and therefore cannot provide a corresponding analysis. Due to the urgency of the surgery and the deteriorated clinical status of most patients, it was not possible to collect such data if not explicitly mentioned in referral letters or patients hospital records. We added this point to the limitations of our study after the first round of Reviewing. Further, we believe that with the GFR we have an objectifiable parameter that can give a reliable statement about the renal function of examined population. Nevertheless, we will consider this constructive suggestion for our upcoming prospective analyses. 

Regarding LVEF, 2.6% of the examined population had LVEF<30% with 2.7% the Non-HD and 1.7% in the HD-group (p = 0.666). This data was added to table 1 after Round 1 Reviewing. We added more details to table 1 as indicated below in red colour. Furthermore, we added more details to the comparison according to GFR stage as indicated below in red colour.

‘’There were no significant differences between groups regarding preoperative left ventricular ejection fraction (LVEF). 7.6% of the patients in the G1 group had LVEF<30% vs. 2.9% in the G2/G3a group (p = 0.285) and 6.7% in the G3b/G4 group (p = 0.942).’’

Table 1. Characteristics of patients surgically treated for Infective Endocarditis.

Variables

All patients

(n=592)

Non-HD group

(n=534)

HD-group

(n=58)

p-value

LVEF <30%

15 (2.6%)

14 (2.7%)

1 (1.7%)

0.666

LVEF 30-50%

108 (26.6%)

92 (26.3%)

16 (28.5%)

LVEF >50%

274 (67.6%)

236 (67.6%)

38 (67.8%)

variables

G1 group

n=143

(26.9%)

G2+G3a

n=228

(42.8%)

p-value

G3b+G4

n=139

(26.1%)

p-value

LVEF <30%

8/105

(7.6%)

5/170

(2.9%)

0.285

5/74

(6.7%)

0.942

LVEF 30-50%

28/105

(26.7%)

44/170

(25.8%)

21/74

(28.3%)

LVEF >50%

71/105

(67.6%)

121/170

(71.2%)

48/74

(64.8%)

Reviewer 3 Report

All the issues raised were properly treated by the authors. 

minor problems